# External negative pressure transiently reduces intravenous pressure and augments the arteriovenous pressure gradient in the affected limb segment

Nigel A. Callender [1,2,3]*, Lars Øivind Høiseth[1,4], Jonny Hisdal[1,2]

**1** Faculty of Medicine, University of Oslo, Oslo, Norway, **2** Department of Vascular Surgery, Oslo University Hospital, Oslo, Norway, **3** Otivio AS, Oslo, Norway, **4** Department of Anaesthesia and Intensive Care, Oslo University Hospital, Oslo, Norway

* nigelac@uio.no

**Data Availability Statement:** The data used is openly available in the Open Source Framework

## Abstract

Recently intermittent negative pressure has emerged as a potential treatment in vascular disease and has similarities with established experimental interventions such as lower body negative pressure. The direct, local influences of either method upon intravascular pressure still require some clarification however, particularly in the immediate moments following onset. We investigated the acute intravascular pressure responses to intermittent cycles of negative pressure in the supine and sitting postures. Fifteen participants (6 female) received intermittent negative pressure cycles (-37 mmHg; 9.5-sec on, 7.5-sec off) upon the lower leg in both postures. Saphenous venous ($n = 15$), and dorsalis pedis artery pressure ($n = 3$) were recorded via pressure catheter, alongside beat-by-beat systemic cardiovascular parameters (heart rate and blood pressure; $n = 15$), from which the arteriovenous pressure gradient was ultimately derived. Negative pressure induced a transient reduction in local intravenous pressure (Supine: 14±3 mmHg to -18±6 mmHg, p<0.001; Sitting: 58±10 mmHg to 41±10 mmHg, $p$<0.001). Rate of venous pressure recovery during the negative pressure plateau phase was faster during sitting, than supine (1.94±0.72 $vs.$ 1.06±0.69 mmHg·sec$^{-1}$; $p = 0.002$). Local intraarterial pressure did not change. External negative pressure readily transmits to the superficial intravenous environment of the leg and transiently augments the arteriovenous pressure gradient. The greatest and most sustained effect was during the supine position. The augmented arteriovenous gradient might briefly produce Poiseuille-dependent haemodynamics before local autoregulatory mechanisms engage. These findings benefit understanding of the immediate *in-vivo* effects of negative pressure upon the local vasculature, and may partly account for the positive clinical effects of intermittent negative pressure treatments in vascular disease.

public repository via the following link: https://osf.io/rnkxd/.

**Funding:** This study was funded by Otivio AS and the Norwegian Research Council (https://www.forskningsradet.no), Grant Ref: 329428. NC is employed as a PhD student by Otivio AS with match-funding from the Norwegian Research Council through the Industrial Ph.D. Scheme/Public Sector Ph.D. Scheme, Ref. 329428. The funders had no role in study design, data collection and analysis, decision to publish, or preparation of the manuscript.

**Competing interests:** NC is employed as a PhD student by Otivio AS with match-funding from the Norwegian Research Council through the Industrial Ph.D. Scheme/Public Sector Ph.D. Scheme, Ref. 329428. This does not alter our adherence to PLOS ONE policies on sharing data and materials.

## Introduction

Intermittent negative pressure (INP) is an emerging treatment option for vascular disease and other disorders of perfusion [1,2] delivering cycles of low-moderate negative pressure surrounding a single limb segment with the aim of influencing local blood flow. This is in contrast to experimental lower body negative pressure (LBNP) which involves sustained negative pressure, applied to a large surface area with the aim of challenging central circulatory homeostasis [3]. Although the systemic effects of LBNP are well documented [3], the mechanisms underlying the positive clinical effect of INP remain unclear, but may relate to oscillations in shear stress—the frictional force between blood and endothelium—within the macro- [4] and micro-vasculature [5,6]. When a small volume of tissue such as a distal limb segment is exposed to negative pressure, a characteristic pattern of blood velocity change is elicited. Immediately following the external pressure reduction blood velocity sharply increases, but rather than being sustained, it soon falls below baseline levels where it remains until the cessation of the external negative pressure [5–8]. While myogenic responses to increased transmural pressure are thought to account for the later reduction in blood velocity [7–9], the initial upstroke may be a result of increased arteriovenous pressure difference and might contribute to any changes in shear stress and by extension, potentially clinical improvements that have been found with INP.

In terms of soft tissue and the vasculature, negative pressure elicits different responses depending on the compartment under investigation. Although negative pressure transmits with limited attenuation to even deep tissue regions [10,11], the vascular network behaves as a functionally separate compartment, continuous between regions under the influence of a higher external pressure [12], in our case the ambient atmosphere, and the lower pressure region enclosed within the chamber. Surprisingly, despite the frequent use of LBNP and knowledge of the vascular network's behaviour as an independent compartment, investigations of intravascular pressure in the region directly enclosed by a negative pressure chamber are few. Aratow et al. [13] measured saphenous venous pressure during LBNP in increments to -30 mmHg, but observed no effect upon intravenous pressure. Conversely, Jacobsen et al. [14] did find a modest reduction in saphenous venous pressure during a -10 mmHg intervention. Methodological differences such as a lower sampling frequency in the former study may account for this discrepancy and highlight potential nuances in the time course of intravenous pressure behaviour, particularly during the early phases of a negative pressure intervention.

Juxtaposed with the leg, the limited investigations of the upper limb show a more consistent pattern of local intravenous pressure reduction, albeit transient in nature and variable in magnitude relative to the external pressure change [8,9,15]. Following a sharp initial decrease, intravenous pressure in the arm progressively returns to near-resting values despite a steadily maintained reduction in external pressure, further confirming the behaviour of the vasculature as a compartment separate to the surrounding tissue [12]. Previous studies have observed this normalisation of intravenous pressure to occur within a minute, driven by ongoing venous filling from continued arterial inflow [8,9,15]. However, the studies described previously are limited by the low number of healthy participants included [8,15] (*n* = 4), [9] (*n* = 3), and more comprehensive description remains warranted. Moreover, whether the lower limb vessels respond similarly to the upper limb remains incompletely defined, but given the reduced distensibility of lower versus upper limb veins in conjunction with disparities in their myogenic responses [7], intravenous pressure change in response to external negative pressure could potentially be attenuated in regions chronically exposed to gravity.

While some limited works have addressed the local pressure changes within the venous system, the arterial network differs in both haemodynamic and mechanical characteristics. Due

to the much lower compliance of arteries compared to veins [16,17], and the arterial system comprising a 'semi-open' valveless conduit [18], it may be assumed that local arterial pressure will resist a response to external negative pressure, however this has not to our knowledge been directly investigated. Furthermore, with relatively limited and sometimes conflicting investigations of local intravascular pressure responses to external negative pressure, the influence upon the arteriovenous pressure gradient, the driving force for perfusion, is also unclear. Considering the characteristic pattern of blood velocity change to sustained [7] or intermittent [5] negative pressure, it may be reasonable to assume that an initial acceleration in velocity could reflect brief Poiseuille-dependent haemodynamics prior to engagement of local autoregulatory mechanisms of vascular resistance [8]. Initially however, simply clarifying the immediate intravascular pressure responses to external negative pressure is crucial to further understand the exact haemodynamic effects being generated regionally within a limb during such interventions, particularly within the first moments following onset. Moreover, from a clinical aspect since INP treatments are largely delivered in a seated posture, knowledge of the interaction between body position and negative pressure on local haemodynamics warrants investigation. Indeed, known differences are present both in venous pressure [19] and lower limb arterial shear rate [20] during upright stances relative to supine postures. This poses the potential for differences in the efficacy of INP between positions, imposed by posturally-derived changes in vascular resistance [21,22].

Specifically focussing on the time course immediately following negative pressure onset, this study aimed to first define the local, lower limb intravenous and intraarterial pressure responses conferred by INP, and any accompanying change in the arteriovenous pressure difference. We hypothesised that a reduction in venous pressure alone would increase the arteriovenous pressure gradient during the negative pressure phase. Second, hypothesising that the influences of INP upon venous pressure would be similar in the supine and sitting postures, we aimed to investigate the intravenous and intraarterial responses to negative pressure between the two positions. These results may further describe the local intravascular pressure changes within the lower limb during experimental perturbations in external pressure and help define the mechanisms underlying the efficacy of INP in a clinical context.

## Methods

### Participants

Fifteen participants were recruited from local clinical staff and students; (9 male, 6 female; age: 29 ± 6.7 y; weight: 70 ± 11 kg; height 1.75 ± 0.08 m). All were free from conditions or medications influencing the cardiovascular system, including venous disease, and refrained from food (>2h), caffeine (>8h), alcohol and heavy exercise (>24h) prior to testing. Following local ethical approval (South-East Norway Regional Ethics Committee: Ref. 2021/323184 & 2022/466181), written, informed consent was provided, and all procedures aligned with the *Declaration of Helsinki*, except for registration in a database. Recruitment to the study commenced 15/01/22 and completed 1/07/23.

### Procedure

Participants attended in either the morning or early afternoon, and following medical screening and basic anthropometric measures, a single leg was instrumented and placed within the negative pressure chamber, sealing just distal to the knee. Following a minimum 20-min period of quiet supine rest, participants undertook 100-sec of intermittent negative pressure (-37 mmHg) in cycles of 9.5-sec negative pressure and 7.5-sec ambient atmospheric pressure. These timings represent the optimal cycle duration to achieve maximum mean blood velocity

as identified during previous investigations [5]. The INP intervention was preceded by a 100-sec baseline recording prior to each trial and was repeated in both a supine and sitting posture. A minimum 10-min normalisation period was observed between any postural change. In the sitting position, knee angle was maintained at ~ 80º flexion. During each condition, three pressure cycles were completed before the initiation of each recording sequence to mitigate the influence of venous stretch during repeated venous filling [23]. Continuous local intravenous ($n$ = 15) and intraarterial ($n$ = 3) pressure measures were recorded, alongside central blood pressure (mmHg), heart rate (bpm) and chamber pressure (mmHg). Laboratory temperature was 24.6 ± 0.9 ºC and ambient atmospheric pressure 751 ± 8 mmHg. All interventions and measurements represent a gauge pressure relative to ambient atmospheric pressure.

## Measures

Height (m; AnthroFlex stadiometer; Henan, China) and weight (kg; Diagnostic XXL; Medel International, Milan, Italy) were recorded using standard anthropometric procedures. Heart rate via 3-lead ECG (Dual Bio Amp; ADInstruments, Dunedin, New Zealand) and beat-by-beat blood pressure via the volume clamp method (Nexfin; BMEYE, Amsterdam, Netherlands) were recorded continuously. Finger blood pressure was assumed to represent central arterial pressure and was recorded from the middle finger of the right hand, which was placed upon an electric warming blanket.

**Negative pressure.** Intermittent negative pressure (-37 mmHg; onset time to 95% plateau pressure = 1.6 sec, offset time = 1.0 sec) was delivered using a commercially available intermittent negative pressure chamber and control unit (FlowOx 2.0; Otivio, Oslo, Norway). A thermoplastic elastomer seal interfaced the leg and chamber approximately 10 cm below the popliteal crease, and all internal padding was removed to prevent movement artefact due to skin contact. Chamber pressure (Stranden Manometer; Stranden Instruments, Ålesund, Norway) was recorded throughout.

**Intravascular pressure.** An intravascular pressure catheter (4.5 cm x 22 g Switch; Vygon, Écouen, France) was inserted retrograde to the direction of flow into the great saphenous vein, level with the medial malleolus. In three participants a separate catheter was inserted into the dorsalis pedis artery (12 cm x 22 g Seldinger arterial catheter; Arrow, Kingston, United Kingdom). Catheter tips were aligned at a similar level within their respective vessels (~2 cm apart in horizontal and longitudinal plane). Catheters were connected to a fluid-filled transducer system (XTrans DPT-9000; Codan, Lensahn, Germany) containing 0.9% NaCl, pressurised to 300 mmHg and running at 3 ml·hr$^{-1}$. Transduced signals were amplified and digitised for continuous recording (BP Amp; ADInstruments, Dunedin, New Zealand). Both catheters were inserted as a sterile procedure, and 1–2 ml Lidocaine 1% infiltrated to the arterial puncture site prior to catheterisation. Pressure transducers were carefully aligned with the catheter tips, adjusted appropriately for each posture, and zeroed to atmospheric pressure prior to each recording bout. All line connections and puncture sites were checked and additionally hermetically sealed with adhesive dressings (Tegaderm; 3M, Neuss, Germany) at any possible air ingress sites. Haemostatic column pressure was measured from the 4$^{th}$ intercostal space to the intravenous catheter tip and calculated using the formula: $\rho \cdot g \cdot h$, where blood density ($\rho$) was taken as 1050 kg · m$^{-3}$. Haemostatic pressure was added to central mean arterial pressure (MAP) for estimation of the arteriovenous pressure difference at the level of the intravenous catheter during the sitting position.

**Skin blood flux.** Skin blood flux was measured using laser doppler fluxometry (Periflux 4001; Perimed, Järfälla, Sweden) during pilot testing in $n$ = 4 participants, from the medial aspect of the lower leg. Signals were sampled using a flat-surfaced probe (Angled 404 probe;

Perimed) affixed to the skin with double-sided adhesive tape, approximately 5–10 cm proximal to the medial malleolus in order to limit skin movement artefacts.

**Signal acquisition.**   All signals were collected using an analogue to digital convertor (PowerLab 16/35; ADInstruments, Dunedin, New Zealand), sampled at 1 KHz and recorded within LabChart software (LabChart Pro 8.0; ADInstruments).

**Analyses.**   Data from Labchart were exported as a.csv file to Microsoft Excel at 50 Hz resolution. Data were visually inspected for movement artefacts and blood pressure calibration sequences, cleaned, and linearly interpolated if appropriate, with four to six complete negative pressure cycles included for each participant. The mean of the final 30-sec of the baseline period was used in analyses. Thereafter, an ensemble-average of the intermittent negative pressure cycles was created for each participant. The onset of negative pressure was identified as the point where a sustained decrease in chamber pressure below -0.15 mmHg occurred, and the offset when an increase in chamber pressure above -36.5 mmHg was sustained.

In addition to the baseline period, four, 1-sec intervals of interest were identified for analysis. These were; the interval corresponding to the nadir in intravenous pressure following the onset of negative chamber pressure defined as the 1-sec period centred upon this point (NP Start); the interval immediately preceding the offset of negative chamber pressure (NP End); the interval following the return of intravenous pressure to baseline after negative chamber pressure ended (AP Start); and finally, the interval at the end of the atmospheric pressure phase, immediately preceding the onset of negative chamber pressure (AP End). The same 1-sec time intervals were used for the comparison of between-posture measures.

Thereafter, data were analysed within RStudio [24], with code included in the supplemental data. Homoscedasticity and normality of variance were determined using the Breusch-Pagan and Shapiro-Wilk tests. Multiple comparisons between time points during the negative pressure cycle were analysed using a one-way, within-subjects ANOVA, with Tukey's *post-hoc* correction. Multiple comparisons against baseline measures alone were *post-hoc* corrected using Dunnett's test. Between-posture analyses were performed using two-tailed, paired-sample Student's *t*-tests. Data are presented as mean ± SD with 95% confidence interval and unless otherwise stated, $p < 0.05$ considered to be statistically significant.

## Results

### Central circulatory variables

No statistically significant difference was present between time points for the following variables in either posture: HR (supine: $F = 0.021$, $p = 0.999$; sitting: $F = 0.009$, $p = 1.000$), central systolic (supine: $F = 0.041$, $p = 0.997$; sitting: $F = 0.062$, $p = 0.993$), central diastolic (supine: $F = 0.059$, $p = 0.993$; sitting: $F = 0.107$, $p = 0.980$), or central mean arterial pressure (supine: $F = 0.081$, $p = 0.988$; sitting: $F = 0.085$, $p = 0.987$) in either posture (all $df = 4$; Table 1). Calculated local arterial pressure (central MAP plus estimated haemostatic pressure) in the sitting position was not significantly different between time points ($F = 0.055$, $p = 0.994$, $df = 4$). Between-posture differences are presented in Table 1.

### Local intravenous pressure responses

External negative pressure induced a rapid, statistically significant decrease in intravenous pressure relative to baseline during both postures (Table 2). A significant difference was identified between time points during the pressure cycle in both the supine ($F = 140.4$, $df = 4$, $p < 0.001$) and sitting position ($F = 8.681$, $df = 4$, $p < 0.001$; *post-hoc* corrected data presented in Table 2). Similarly, a significant difference in the absolute arteriovenous difference was found

**Table 1. Influence of the external negative pressure cycle upon cardiovascular variables at key time points.**

|  | Baseline | NP Start | NP End | AP Start | AP End |
|---|---|---|---|---|---|
| **Supine** | | | | | |
| **HR (bpm)** | 58 ± 10 * | 57 ± 10 * | 57 ± 10 * | 57 ± 10 * | 57 ± 10 * |
| *95% CI* | *52–63* | *52–62* | *52–62* | *51–62* | *52–62* |
| **Central MAP (mmHg)** | 84 ± 11 [a] | 83 ± 9 * | 84 ± 9 * | 84 ± 10 * | 85 ± 10 [a] |
| *95% CI* | *79–89* | *79–88* | *79–89* | *79–89* | *80–90* |
| **SBP (mmHg)** | 116 ± 16 | 116 ± 14 [b] | 117 ± 14 [c] | 116 ± 14 [a] | 118 ± 15 |
| *95% CI* | *108–124* | *109–123* | *110–124* | *109–123* | *110–126* |
| **DBP (mmHg)** | 65 ± 8 * | 64 ± 7 * | 64 ± 7 * | 65 ± 8 * | 65 ± 8 [d] |
| *95% CI* | *61–69* | *61–68* | *61–68* | *61–68* | *61–69* |
| **Local MAP (mmHg)** | NA | NA | NA | NA | NA |
| *95% CI* | | | | | |
| **Sitting** | | | | | |
| **HR (bpm)** | 69 ± 15 | 69 ± 12 | 70 ± 13 | 69 ± 12 | 69 ± 13 |
| *95% CI* | *62–77* | *63–75* | *63–77* | *63–76* | *62–76* |
| **Central MAP (mmHg)** | 91 ± 12 | 91 ± 11 | 91 ± 12 | 92 ± 12 | 90 ± 11 |
| *95% CI* | *85–98* | *85–97* | *85–97* | *86–99* | *84–96* |
| **SBP (mmHg)** | 122 ± 16 | 121 ± 15 | 122 ± 15 | 123 ± 16 | 120 ± 15 |
| *95% CI* | *114–131* | *114–129* | *114–129* | *115–131* | *113–128* |
| **DBP (mmHg)** | 73 ± 10 | 72 ± 10 | 73 ± 10 | 74 ± 10 | 71 ± 10 |
| *95% CI* | *67–78* | *68–77* | *68–78* | *68–79* | *66–76* |
| **Local MAP (mmHg)** | 147 ± 15 | 147 ± 14 | 147 ± 14 | 148 ± 15 | 146 ± 14 |
| *95% CI* | *139–155* | *140–154* | *140–155* | *141–156* | *139–153* |

All data are mean ± SD. NP: Negative pressure phase, AP: Atmospheric pressure phase, HR: Heart rate (beats per minute), central MAP: Central mean arterial pressure (mmHg), SBP: Central systolic blood pressure (mmHg), DBP: Central diastolic blood pressure (mmHg), Local MAP: Estimated local MAP, the product of central MAP plus estimated haemostatic pressure (mmHg), note in the supine posture this is assumed to be similar to central MAP. NP Start: 1-sec period encompassing the nadir of venous pressure following the onset of external negative pressure, NP End: Final 1-sec of external negative pressure plateau phase, AP Start: 1-sec period following return of intravenous pressure to baseline pressure. AP End: Final 1-sec of atmospheric pressure phase. $n = 14$ for central MAP, SBP and DBP during the Baseline period in sitting.

* denotes significant difference between postures (p<0.001).

[a] denotes sig difference between postures (p = 0.002).

[b] denotes significant difference between postures (p = 0.010).

[c] denotes significant difference between postures (p = 0.017),

[d] denotes significant difference between postures (p = 0.001).

in the supine ($F = 31.62$, $df = 4$, $p<0.001$) and sitting positions ($F = 4.374$, $df = 4$, $p = 0.003$; *post-hoc* corrected data presented in Table 2).

In addition to the longer 1-sec intervals investigated above, the single-point (0.1-sec interval) minimum intravenous pressure was identified in both postures during the external negative pressure phase. This maximum decrease in intravenous pressure was found to represent 87 ± 10% of that applied by the chamber during the supine position, and 50 ± 18% during sitting and with the difference between postures found to be statistically significant ($t = 7.665$, $p<0.001$). Over the final 7-sec interval during the plateau phase of negative pressure (thus ensuring all data represented the venous pressure recovery phase only), intravenous pressure recovered towards baseline more rapidly in the sitting position (rate of pressure change: 1.94 ± 0.72 mmHg·sec$^{-1}$) compared to supine (rate of pressure change: 1.06 ± 0.69 mmHg·sec$^{-1}$; $t = -3.895$, $p = 0.002$).

**Table 2. Influence of the external negative pressure cycle upon intravenous pressure and arteriovenous pressure difference at key time points.**

| | Baseline (mmHg) | NP Start (mmHg) | NP End (mmHg) | AP Start (mmHg) | AP End (mmHg) |
|---|---|---|---|---|---|
| **Venous Pressure** | | | | | |
| **Supine** | | | | | |
| Mean ± SD | 14 ± 3 | -18 ± 6 [*#†] | -11 ± 7 [*#†a] | 18 ± 5 | 15 ± 5 |
| CI 95% | 12–15 | -21--15 | -15--8 | 15–21 | 12–18 |
| **Sitting** | | | | | |
| Mean ± SD | 58 ± 10 | 41 ± 10 [*] | 54 ± 10 [b] | 59 ± 10 [$] | 58 ± 10 [$] |
| 95% CI | 53–63 | 36–46 | 50–59 | 54–64 | 53–63 |
| Supine vs. Sitting | <0.001 | < 0.001 | <0.001 | <0.001 | <0.001 |
| t-statistic | -16.48 | -19.175 | -21.789 | -15.232 | -15.628 |
| **Arteriovenous Pressure Difference** | | | | | |
| **Supine** | | | | | |
| Mean ± SD | 70 ± 12 | 101 ± 10 [*#†] | 95 ± 11 [*#†] | 66 ± 11 | 70 ± 12 |
| 95% CI | 65–76 | 96–106 | 90–101 | 61–72 | 64–76 |
| **Sitting** | | | | | |
| Mean ± SD | 90 ± 13 | 106 ± 15 [cde] | 93 ± 14 | 89 ± 13 | 88 ± 12 |
| 95% CI | 83–97 | 98–113 | 86–100 | 83–96 | 82–94 |
| Supine vs. Sitting | <0.001 | 0.219 | 0.449 | <0.001 | <0.001 |
| t-statistic | -5.842 | -1.287 | 0.779 | -8.280 | -6.140 |

All values are mean ± SD, mmHg. Venous pressure: The absolute venous pressure recorded. Arteriovenous pressure difference: Estimated during supine as: (central MAP—venous pressure), and during sitting as: ((central MAP + haemostatic pressure)—venous pressure). NP Start: 1-sec period encompassing the nadir of venous pressure following the onset of external negative pressure, NP End: Final 1-sec of external negative pressure plateau phase, AP Start: 1-sec period following return of intravenous pressure to baseline pressure. AP End: Final 1-sec of atmospheric pressure phase. Supine vs. sitting: p-value of between-posture comparisons. t-statistic: T-test statistic for between-posture comparisons. n = 14 for arteriovenous pressure difference, central MAP, SBP and DBP during the Baseline period in sitting.

[*] denotes significant difference to baseline (p<0.001).

[#] denotes significant difference to AP Start (p<0.001),

[†] denotes significant difference to AP End (p<0.001),

[$] denotes significant difference to NP Start (p<0.001),

[a] denotes significant difference to NP Start (p = 0.022).

[b] denotes significant different to NP Start (p = 0.004),

[c] denotes significant different to baseline (p = 0.021),

[d] denotes significant different to AP start (p = 0.011),

[e] denotes significant different to AP End (p = 0.004).

Absolute venous pressures at all timepoints during baseline and the external pressure cycle were significantly different between postures (all $p$ <0.001; Table 2). Temporal changes during both postures are presented in Fig 1.

## Local arterial pressure

In order to ascertain whether central arterial pressure with any corresponding correction for haemostatic pressure would accurately represent local intraarterial pressure, direct local arterial pressure measurement was undertaken in three participants. The pattern of dorsalis pedis pressure change (within the area directly influenced by negative pressure) was found to reflect central MAP (Fig 2). No apparent effect of external negative pressure upon local MAP was observed in either the supine (Negative pressure phase: 123 ± 10 mmHg, atmospheric pressure phase: 123 ± 11 mmHg) or sitting posture (Negative pressure phase: 189 ± 11 mmHg,

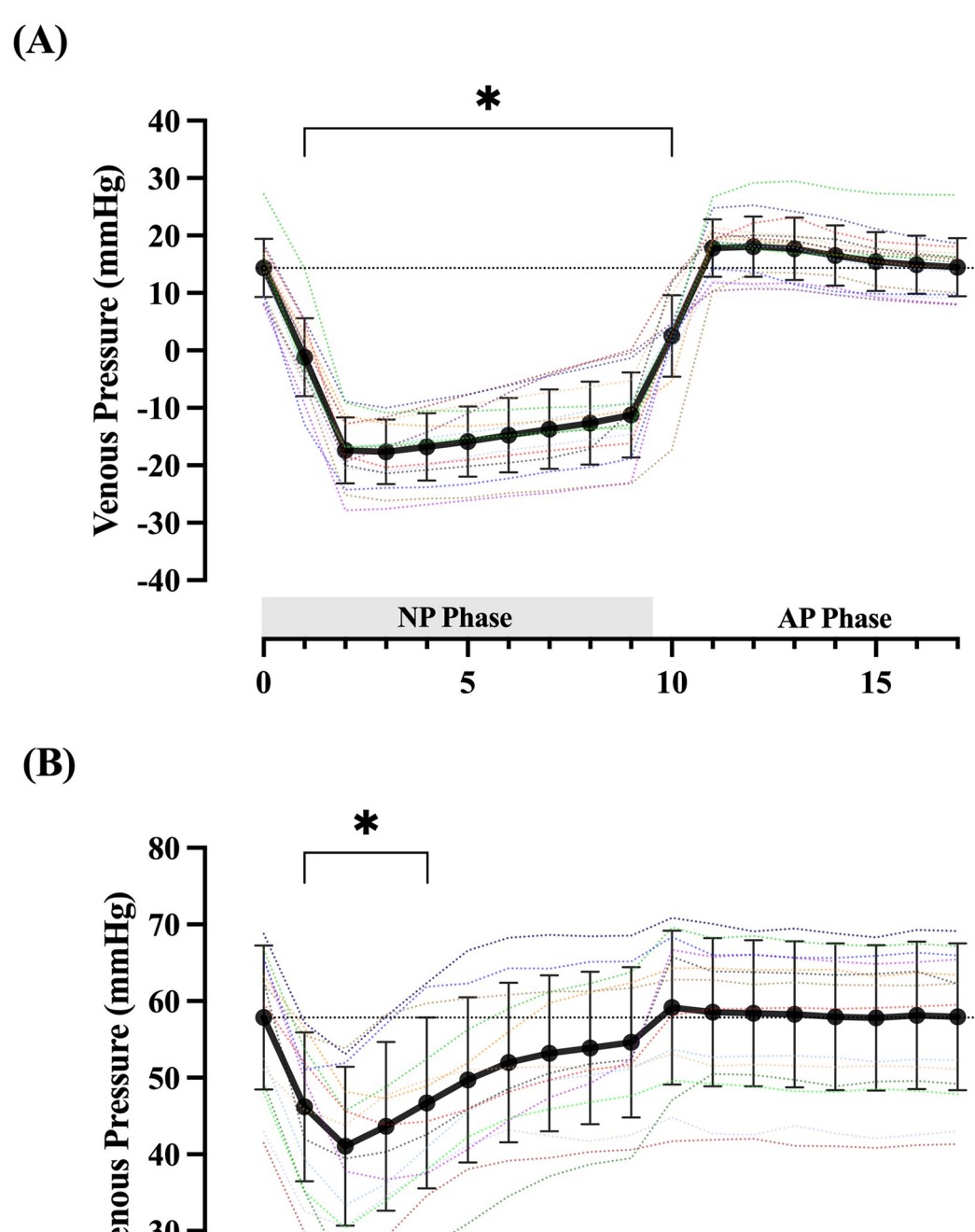

**Fig 1. Time course of venous pressure change during the negative pressure cycle.** Venous pressure response to negative external pressure in the supine (Panel A) and sitting (Panel B) postures. Data represent the mean ± SD of all participants sampled at each whole second from 0–17 sec (Solid black line) with individual participants plotted as dotted lines. Dotted horizontal line = baseline intravascular pressure (group mean of 30-sec baseline period). Shaded horizontal bar = Negative chamber pressure phase (NP Phase), 0–9.5-sec. AP Phase = Atmospheric pressure phase, 9.5–17-sec. * = significantly different from baseline venous pressure ($p < 0.001$).

Supine                                                    Sitting

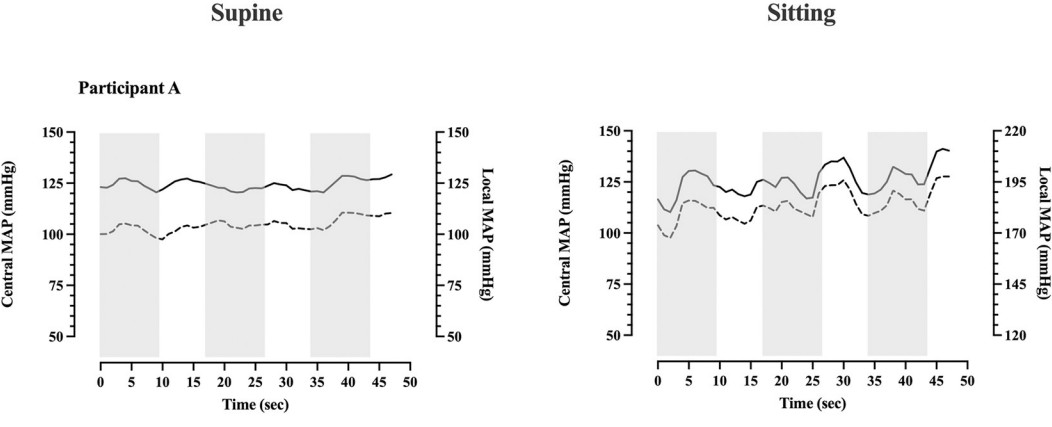

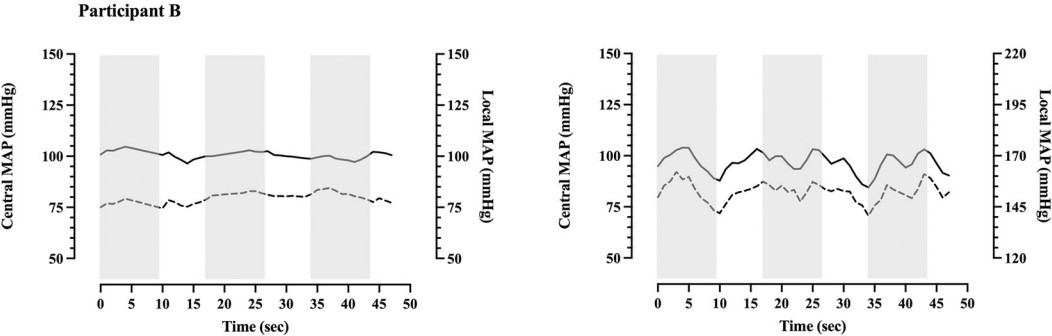

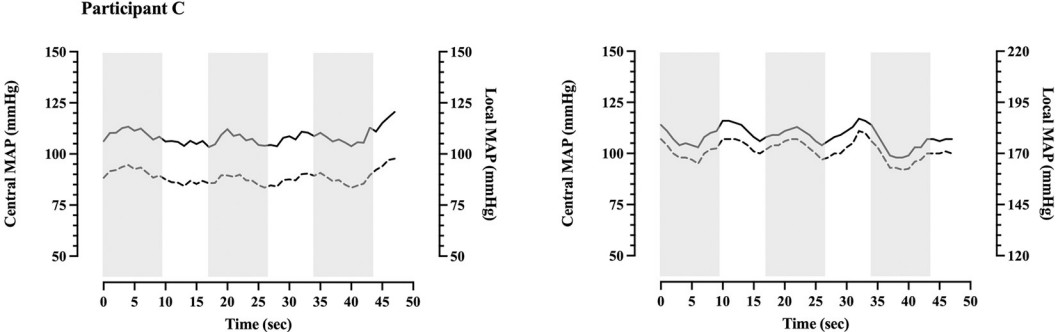

**Fig 2. Central and local intraarterial pressure traces during intermittent cycles of external negative pressure.** Local intraarterial MAP closely reflects central MAP and appears uninfluenced by local external negative pressure. Data represent continuous beat-by-beat central (Nexfin derived) and local (dorsalis pedis arterial catheter) MAP for three individual participants. Three complete cycles are presented. Shaded areas indicate periods of external negative pressure.

atmospheric pressure phase: 190 ± 10 mmHg; pressures represent the mean of the complete pressure interval). Given its invasive nature and these initial findings, arterial catheterisation was discontinued for the remaining participants and central MAP plus correction for haemostatic pressure was deemed to acceptably estimate local arterial pressure.

## Discussion

In the lower limb, we aimed to describe the immediate changes in intravenous and intraarterial pressure in response to short, intermittent cycles of negative pressure at -37 mmHg and additionally, what effects posture may have upon these changes. Cycles of external negative pressure elicited a transient decrease in local intravenous pressure during both postures, although the effect was diminished in the sitting, relative to supine position. These effects were limited to the venous network alone, with no apparent change in central, or crucially, local intraarterial pressure. This resulted in a transiently increased arteriovenous pressure difference in the region exposed to negative pressure.

### Intravenous pressure responses to negative pressure and the influence of posture

Baseline intravenous pressures in the supine (14 ± 3 mmHg) and sitting (58 ± 10 mmHg) positions were in line with previously reported values [25], the difference accounted for by the greater haemostatic column present in the more upright posture. During supine, external negative pressure induced a fall in intravenous pressure from 14 ± 3 to -18 ± 6 mmHg (Table 2). At its maximum, the absolute decrease in intravenous pressure attained 87 ± 10% of the pressure change within the enclosing chamber. The limited previous works examining saphenous venous pressure under similar, but steady-state conditions have been somewhat inconsistent in their findings, reporting either no influence of LBNP [13], or up to ~70% transmission of the external pressure change [14]. In particular, the lack of any influence from LBNP noted by Aratow *et al.* (10) may be accounted for by a long sampling interval which, due to ongoing venous filling, could miss a transient drop in intravenous pressure (e.g. Fig 1). Their findings, alongside our own data, highlight the time dependence of haemodynamic changes under such conditions, even when the initial intravenous pressure is low and inflowing blood can readily be accommodated within the vessel. Our findings in the supine position appear broadly consistent with that of Jacobsen et al. (11), and also selected works using similar techniques in the upper limb [9,15]. The similarity between our findings and previous observations in the arm also suggest little difference in the time course or magnitude of pressure change between veins accustomed to chronic gravitational stress and those in the upper limb [26,27], however this warrants direct comparison in future studies.

In contrast to the supine position, intravenous pressure fell from 58 ± 10 to 41 ± 10 mmHg during sitting, attaining only 50 ± 18% of the absolute change in external chamber pressure. Accompanying this, the rate of intravenous pressure recovery towards baseline was almost twice as fast in the more upright position (1.94 ± 0.72 mmHg· sec$^{-1}$ vs. 1.06 ± 0.69 mmHg· sec$^{-1}$; p = 0.002). This is despite a reduced arterial inflow typically being expected during sitting due to the effects of increased vascular resistance in this position [21,22,28]. The difference in behaviour noted between postures, both in the rate of intravenous pressure rise following negative pressure onset and the altered magnitude of intravenous pressure change accompanying the intervention, we attribute to the greater baseline transmural pressure generated by sitting, and a potential shift from conditions of higher to lower venous compliance [29]. However, future work should investigate the interaction between posture, venous flow, and vascular

compliance during negative pressure before conclusions on the exact mechanism of this difference can be made.

### Influences upon the arteriovenous gradient and potential implications for blood velocity

When the atmospheric pressure surrounding a limb is abruptly reduced, conduit artery mean blood velocity [4,5,7,8] and skin blood flux [5,6] undergo a rapid, but transient acceleration which typically lasts ~2–4 sec. Thereafter, blood velocity peaks, then decelerates to remain below baseline levels until the external pressure reduction ceases, as is illustrated within pilot data included in Fig 3. The arrest of acceleration, and subsequent reduction in blood velocity owes, at least in part, to local autoregulatory responses [5,8] which rapidly respond to altered transmural pressure [30]. The preceding acceleration in blood velocity has been postulated to occur from passive distension of the arterial/arteriolar network, or to a negative pressure developing within the veins [8].

In our investigations, direct arterial pressure measurements within the region exposed to negative pressure were undertaken in three participants. These demonstrated no influence of

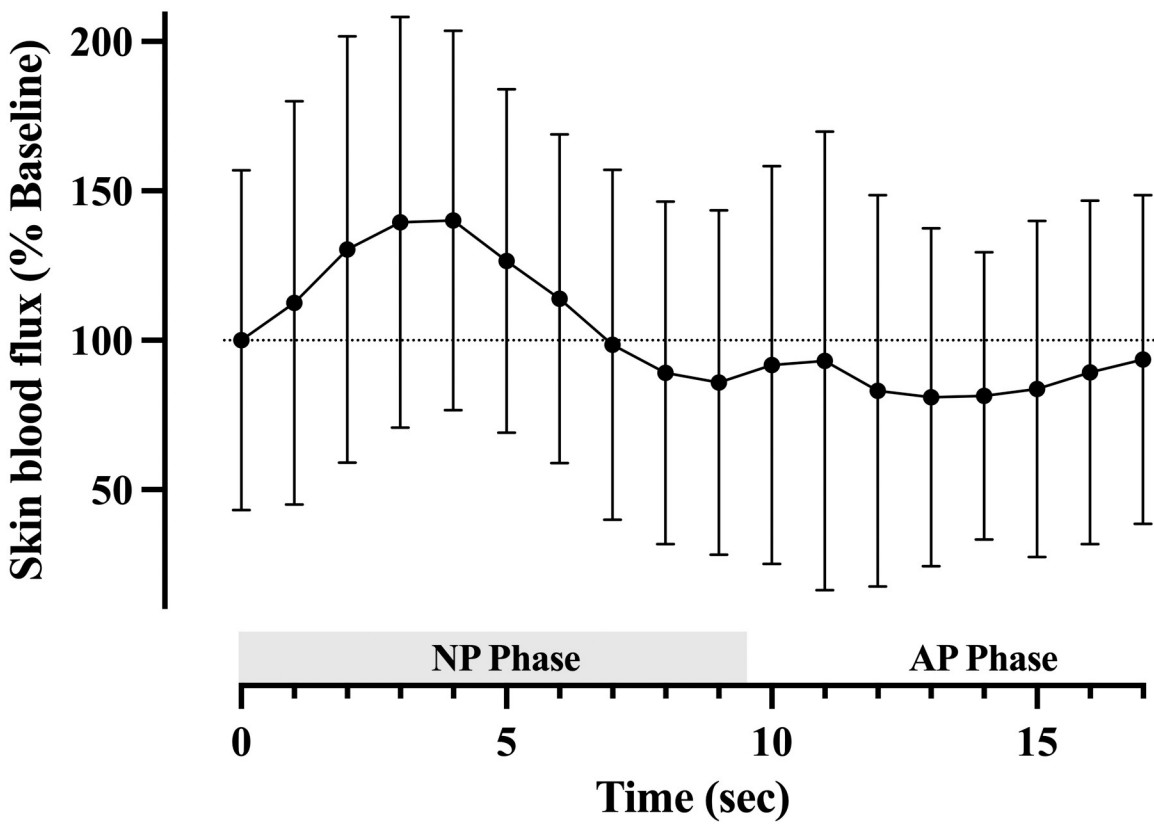

**Fig 3. Changes in skin blood flux during the negative pressure cycle.** Laser doppler flux recorded upon skin superior to the medial malleolus during the negative pressure cycle during a pilot experiment. Note the initial acceleration in skin blood flux, followed by a subsequent deceleration below baseline during the 2–3 seconds preceding restoration of ambient atmospheric pressure. Four to six pressure cycles were aggregated for each of $n = 4$ participants in a supine position. Data represent the mean ± SD of each whole second from 1–17 sec and are expressed as percent change relative to baseline flux. Time 0 = mean of 1-sec period prior to onset of negative pressure. Dotted horizontal line = baseline skin blood flux. Shaded horizontal bar = Negative chamber pressure phase (NP Phase), 0–9.5-sec. AP Phase = Atmospheric pressure phase, 9.5–17-sec.

external negative pressure of -37 mmHg upon the local conducting arteries. This is probably the result of two factors: First, the arterial system is a valveless conduit with a reservoir (the left ventricle) at one end, which permits rapid, unopposed inflow of blood. This largely unresisted inflow would offset any potential pressure reduction occurring in response to an increase in distal arterial volume. Second, the pressures required to distend these proportionally more muscular, and therefore less compliant arterial vessels [17] appear to be far in excess of those delivered here. Indeed, work examining the intraarterial pressure thresholds at which conducting arterial diameter is altered, did not observe an effect until additional distending pressure exceeded ~200 mmHg [31]. These findings suggest that any haemodynamic effects are probably the result of a reduction in venous pressure, rather than arterial distention. These findings also increase our confidence that the use of systemic arterial pressure plus any correction for haemostatic pressure provides a valid representation of local arterial pressure in regions exposed external negative pressure.

In both postures, intravenous pressure fell simultaneous to chamber pressure, with the greatest decrease directly following negative pressure onset (Fig 1). While ongoing arterial inflow eventually equilibrates the local intravenous pressure with downstream venous pressure, since arterial pressure remains unaffected, the local arteriovenous pressure gradient is increased during the intervening period (Table 2). These findings are consistent with investigations by Lott et al. [8], and lend support to a mechanism where during ideal conditions, an augmented arteriovenous gradient may for brief moments, drive the increase in blood velocity in a Poiseuille-dependent manner. Thereafter, in response to the increase in transmural pressure, engagement of local autoregulatory mechanisms induces a rapid increase in local vascular resistance [7,8,26], which alone, or in combination with the diminishing arteriovenous pressure gradient may account for the subsequent fall in blood velocity.

## Clinical perspectives

Although our study involved only healthy participants, our findings allow some speculation as to the possible mechanisms underpinning any positive clinical effects of INP. Periodic increases in the local arteriovenous gradient could lead to improved vascular responses via shear-mediated improvements in endothelial function [4]. Indeed, oscillatory changes to limb haemodynamics in response to INP have been demonstrated to improve upper limb flow mediated dilation, a biomarker of endothelial function, among healthy volunteers [4]. Whether similar effects occur among patients with arterial disease remains to be established. However, INP has been shown to induce haemodynamic oscillations even among patients with significant vascular disease [6,32,33], where shear related effects were postulated as the mechanism underpinning the observed clinical improvements [1,32].

Moreover, under certain conditions it is possible that the increased arteriovenous pressure gradient may benefit local perfusion, independent of any shear-related effects. In patients with significant arterial disease, the overriding need to maintain tissue oxygen delivery leads to attenuation of local vascular autoregulatory responses to prioritise blood flow [34]. This results in a more pressure-dependent circulation, which would render these regions more susceptible to haemodynamic changes resulting solely from alterations in vascular pressure gradient, rather than reliant upon changes to endothelial function. Indeed, patient groups such as those with intermittent claudication [1,35] and critical ischaemia [36] who both may exhibit these vascular characteristics have been shown to benefit from courses of INP treatment. Future work should aim to define any haemodynamic differences between healthy volunteers, as investigated here, and patients with vascular disease.

## Limitations

Our data only represent the vascular pressure responses within the superficial vessels thus we cannot exclude an attenuated effect of external negative pressure upon the deep venous network. However, with external pressure known to transmit relatively unchanged to the deep compartments of the leg [11], it is likely that similar patterns occur in the deep veins. Moreover, although a highly consistent pattern of blood velocity change is observed during negative pressure, we did not directly measure arterial or venous blood velocity in this study. Therefore, our findings only allow us to specifically comment on the pressure changes occurring in response to INP. Future work should include additional haemodynamic measurements to provide a more complete picture of the responses to negative pressure.

We attribute the attenuated response during sitting to reduced venous compliance in this position, however, we must acknowledge the potential for catheter-induced valvular disruption permitting some limited venous back filling. However, in the saphenous vein, valve intervals are such that with the length of catheter used here, few, if any valves may have been disrupted [37]. Furthermore, the more rapid pattern of pressure increase in the upright position was highly consistent between participants, suggesting this phenomenon to more likely represent an effect of the posture itself.

Due to the invasive nature of the investigation, the arterial measurements obtained here were limited to a very small sample ($n = 3$) which in turn limits the generalisability of our results even among healthy participants. However, the local arterial measures reliably reflected variations in MAP throughout the experiments in tandem with the non-invasive measures (Fig 1). Although a constant systematically greater local MAP is reported here (~ 20mmHg in both supine and sitting), this is accounted for by the known differences in arterial pressure between the central and distal circulation [38], and the potential discrepancy produced by the volume clamp method, where values are an estimation of brachial pressure rather than direct measurements. Similarly, the arterial pressure measurements we obtained were sensitive to changes in limb position, and mechanical perturbation (e.g. tapping or flushing) of the artery itself, therefore, we are confident that our local arterial pressure measures would detect any relevant influence of an external pressure change.

## Conclusion

We examined the immediate effects of intermittent reductions in the atmospheric pressure surrounding a limb upon the local intravascular pressure, in both the supine and sitting postures. We noted effective transmission of external negative pressure to the intravenous environment, but no influence upon intraarterial pressure, transiently increasing the arteriovenous pressure gradient. This effect upon intravenous pressure was greatest in the supine posture, but substantially attenuated in a sitting position. These results extend previous work in the upper limb, confirming a similar pattern of intravenous pressure response in the leg. Moreover, we suggest the augmented arteriovenous pressure gradient may explain the initial increase in blood velocity typically observed at the onset of negative pressure, and that any clinical effects INP therapies might be mediated through the shear stress changes.

## Acknowledgments

We wish to thank our participants for giving their time to contribute to the study. We also wish to thank Dr. Iacob Mathiesen for many discussions relating to intermittent negative pressure therapy, and Dr. Sole Lindvåg Lie, Ms. Aura Koistinaho and Dr. William Morton for assistance during testing and providing helpful comments on the manuscript.

## Author Contributions

**Conceptualization:** Nigel A. Callender, Jonny Hisdal.

**Data curation:** Nigel A. Callender.

**Formal analysis:** Nigel A. Callender, Lars Øivind Høiseth.

**Funding acquisition:** Jonny Hisdal.

**Investigation:** Nigel A. Callender, Lars Øivind Høiseth, Jonny Hisdal.

**Methodology:** Nigel A. Callender, Lars Øivind Høiseth, Jonny Hisdal.

**Project administration:** Nigel A. Callender.

**Resources:** Nigel A. Callender, Jonny Hisdal.

**Software:** Jonny Hisdal.

**Supervision:** Lars Øivind Høiseth, Jonny Hisdal.

**Validation:** Nigel A. Callender, Lars Øivind Høiseth.

**Visualization:** Nigel A. Callender.

**Writing – original draft:** Nigel A. Callender.

**Writing – review & editing:** Nigel A. Callender, Lars Øivind Høiseth, Jonny Hisdal.

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
