## [Decision Letter · Decision Letter 0]

13 May 2024

PONE-D-24-09979External negative pressure transiently reduces intravenous pressure and augments the arteriovenous pressure gradient in the affected limb segmentPLOS ONE

Dear Dr. Callender,

Thank you for submitting your manuscript to PLOS ONE. After careful consideration, we feel that it has merit but does not fully meet PLOS ONE’s publication criteria as it currently stands. Therefore, we invite you to submit a revised version of the manuscript that addresses the points raised during the review process.

We look forward to receiving your revised manuscript.

Kind regards,

Haipeng Liu

Academic Editor

PLOS ONE

https://journals.plos.org/plosone/s/file?id=ba62/PLOSOne_formatting_sample_title_authors_affiliations.pdf"

2.Thank you for stating the following financial disclosure: 

"This study was funded by Otivio AS and the Norwegian Research Council (https://www.forskningsradet.no), Grant Ref: 329428. NC is employed as a PhD student by Otivio AS with match-funding from the Norwegian Research Council through the Industrial Ph.D. Scheme/Public Sector Ph.D. Scheme, Ref. 329428." 

3.Thank you for stating the following in the Competing Interests section: 

"NC is employed as a PhD student by Otivio AS with match-funding from the Norwegian Research Council through the Industrial Ph.D. Scheme/Public Sector Ph.D. Scheme, Ref. 329428."

Reviewers' comments:

Reviewer's Responses to Questions

**Comments to the Author**

1. Is the manuscript technically sound, and do the data support the conclusions?

Reviewer #1: Yes

Reviewer #2: Yes

Reviewer #3: Yes

2. Has the statistical analysis been performed appropriately and rigorously? 

Reviewer #1: Yes

Reviewer #2: Yes

Reviewer #3: Yes

3. Have the authors made all data underlying the findings in their manuscript fully available?

Reviewer #1: No

Reviewer #2: Yes

Reviewer #3: Yes

4. Is the manuscript presented in an intelligible fashion and written in standard English?

Reviewer #1: Yes

Reviewer #2: Yes

Reviewer #3: Yes

5. Review Comments to the Author

Reviewer #1: ”Saphenous venous (n = 15), and dorsalis pedis artery pressure (n = 3) were recorded via pressure

catheter, alongside beat-by-beat systemic cardiovascular parameters.“ Can you clarify the precise meaning of "N"?

If "N" represents the number of different measurement points, then it seems that the measurement errors for arteriovenous pressure are not at the same level.

Reviewer #2: Callender and colleagues sought to determine the intravascular pressure responses to intermittent negative pressure (INP) in the supine and seated postures. They observed that INP induced a transient decrease in leg intravenous, but not intraarterial pressure in both postures, though the effect was more pronounced in the supine position. Overall, the study is well written and the findings are interesting. I have only a few suggestions for the authors to consider.

The authors should add the skin blood flux measurements in the methods section and describe the results accordingly (excluding its reference from the introduction line 66).

Lines 121-127: Please add the author’s hypotheses.

Lines 144-146: What is the rationale for this intermittent protocol (i.e., pressure applied and cycle time)? Was it based on previous studies? If so, please provide the references.

The data from the Supplementary Figure is important and should be included in the main manuscript.

Figure 2: It is recommended to add a panel with the group average and individual data of the nadir responses. This will help the readers to better visualize the interindividual variability.

Reviewer #3: This manuscript describes a study investigating the effect of an intermittent negative pressure treatment on the intravenous and intraarterial pressure in a cohort of volunteers. The authors concluded that the treatment resulted in a significant decrease in the intravenous pressure and therefore in an increase in the pressure gradient both in the supine and in the sitting position. The conclusion is well supported by the data. The study is well conducted, the methodology is sound, the statistical analysis has been performed appropriately and rigorously and the manuscript is overall well written. All data underlying the findings have been made fully available by the authors. Minor proofreading is needed.

Line 59 to 61 - The authors wrote: Although the systemic effects of LBNP are well documented (3), the mechanisms underlying the positive clinical effect of INP remain unclear, but may relate to oscillations in shear stress within the macro (4) and micro - vasculature (5,6) Since this is the first time the acronym LBPN appears in the text, you should spell it out.

Lines 286: The authors wrote: Temporal changes during the during both postures are presented in Figure 1. I think this should be Temporal changes during both postures are presented in Figure 1

Line 469 to 472 - The authors wrote: Similarly, the arterial pressure measurements we obtained were sensitive to changes in limb position, and mechanical perturbation (e.g. tapping or flushing) of the artery itself, therefore, we are therefore confident that our local arterial pressure measures would detect any relevant influence of an external pressure change. The word therefore is spelled twice. Please correct into Similarly, the arterial pressure measurements we obtained were sensitive to changes in limb position, and mechanical perturbation (e.g. tapping or flushing) of the artery itself, therefore, we are confident that our local arterial pressure measures would detect any relevant influence of an external pressure change

This manuscript describes a study investigating the effect of an intermittent negative pressure treatment on the intravenous and intraarterial pressure in a cohort of volunteers. The authors concluded that the treatment resulted in a significant decrease in the intravenous pressure and therefore in an increase in the pressure gradient both in the supine and in the sitting position. The conclusion is well supported by the data. The study is well conducted, the methodology is sound, the statistical analysis has been performed appropriately and rigorously and the manuscript is overall well written. All data underlying the findings have been made fully available by the authors. Minor proofreading is needed

6. PLOS authors have the option to publish the peer review history of their article (what does this mean?). If published, this will include your full peer review and any attached files.

Reviewer #1: No

Reviewer #2: No

Reviewer #3: No

---

## [Author Response · Author response to Decision Letter 0]

27 May 2024

Response to reviewers 21-5-24

External negative pressure transiently reduces intravenous pressure and augments the arteriovenous pressure gradient in the affected limb segment.

Many thanks for the kind comments and assistance with improving our manuscript. Responses to the individual points raised are below, with the relevant changes tracked within the updated manuscript.

Reviewer 1

”Saphenous venous (n = 15), and dorsalis pedis artery pressure (n = 3) were recorded via pressure catheter, alongside beat-by-beat systemic cardiovascular parameters.“ Can you clarify the precise meaning of "N"? If "N" represents the number of different measurement points, then it seems that the measurement errors for arteriovenous pressure are not at the same level.¨

Thank you for this request for clarification, these are indeed the number of observations obtained from our sample for saphenous intravenous pressure (n = 15), and intraarterial pressure from the dorsalis pedis artery (n = 3). We have added the further clarification that the arterial pressure measurements obtained from the beat-by-beat finger cuff measurements (Nexfin) were used for the estimation of local mean arterial pressure (n = 15), as is clearly described in the main method section, but due to limited word count, not included in the abstract. As should now be clearer in the abstract, both venous pressure and the estimates of local MAP are of equal size, n = 15. (Line 30-31).

Reviewer 2

The authors should add the skin blood flux measurements in the methods section and describe the results accordingly (excluding its reference from the introduction line 66).

As requested, details of the skin blood flux measurements carried out during pilot testing have been added to the methods section (Line 216-221), and the reference in the Introduction removed.

Lines 121-127: Please add the author’s hypotheses.

Thank you for this suggestion, our initial hypotheses have been added to the final section of the Introduction (Lines 135-139)

Lines 144-146: What is the rationale for this intermittent protocol (i.e., pressure applied and cycle time)? Was it based on previous studies? If so, please provide the references.

Following an initial transient increase, negative pressure, if sustained, induces a reduction in blood flux or blood velocity, often to levels below the pre-intervention baseline (e.g. (Lott et al., 2009; Sundby et al., 2016). For obvious reasons this effect sustained negative pressure is likely to be detrimental to local perfusion if considered from the perspective of being a clinical treatment strategy. Therefore, by applying negative pressure as intermittent cycles, the periods where blood flow is increased during the negative pressure phases can be maximised, and negative pressure only sustained until the point where blood flow is expected to fall below baseline levels. 

Previous work (Sundby et al., 2016) has identified cycles of 10-sec negative pressure, 7-sec atmospheric pressure to be the optimal durations to capitalise on the effect of augmenting blood flow, while returning to atmospheric pressure only for the duration required to offset the vasoconstrictive effect of the negative pressure phase, thus ensuring the greatest mean blood flow over a full intermittent negative pressure cycle. As requested, we have included a brief justification for this and the appropriate reference within the Method section (Line 162-163).

Lott, M. E. J., Hogeman, C., Herr, M., Bhagat, M., Kunselman, A., & Sinoway, L. I. (2009). Vasoconstrictor responses in the upper and lower limbs to increases in transmural pressure. Journal of Applied Physiology, 106(1), 302–310. https://doi.org/10.1152/japplphysiol.90449.2008

Sundby, Ø. H., Høiseth, L. Ø., Mathiesen, I., Jørgensen, J. J., Weedon-Fekjaer, H., & Hisdal, J. (2016). Application of intermittent negative pressure on the lower extremity and its effect on macro- and microcirculation in the foot of healthy volunteers. Physiological Reports, 4(17), e12911. https://doi.org/10.14814/phy2.12911

The data from the Supplementary Figure is important and should be included in the main manuscript.

Thank you for this point. We agree and so the figure (Fig. 3) detailing the invasive arterial pressure measurements has been returned to the main manuscript.

Figure 2: It is recommended to add a panel with the group average and individual data of the nadir responses. This will help the readers to better visualize the interindividual variability.

Thank you for this suggestion, we agree that this is important particularly given the relatively limited sample size. We have included the complete plots for each individual participant in the figure (n = 15), with the mean and SD for the group remaining highlighted as previously. We believe this now conveys much richer information about the dataset, in addition to addressing your recommendation.

Reviewer 3

Line 59 to 61 - The authors wrote: Although the systemic effects of LBNP are well documented (3), the mechanisms underlying the positive clinical effect of INP remain unclear, but may relate to oscillations in shear stress within the macro (4) and micro - vasculature (5,6) Since this is the first time the acronym LBPN appears in the text, you should spell it out.

Thank you for spotting this oversight. This has been added (Line 67).

Lines 286: The authors wrote: Temporal changes during the during both postures are presented in Figure 1. I think this should be Temporal changes during both postures are presented in Figure 1.

Again, thank you for spotting the error. This has been corrected (Line 334).

Line 469 to 472 - The authors wrote: Similarly, the arterial pressure measurements we obtained were sensitive to changes in limb position, and mechanical perturbation (e.g. tapping or flushing) of the artery itself, therefore, we are therefore confident that our local arterial pressure measures would detect any relevant influence of an external pressure change. The word therefore is spelled twice. Please correct into Similarly, the arterial pressure measurements we obtained were sensitive to changes in limb position, and mechanical perturbation (e.g. tapping or flushing) of the artery itself, therefore, we are confident that our local arterial pressure measures would detect any relevant influence of an external pressure change

Corrected (Line 617). Thank you again.

---

## [Decision Letter · Decision Letter 1]

28 Oct 2024

PONE-D-24-09979R1External negative pressure transiently reduces intravenous pressure and augments the arteriovenous pressure gradient in the affected limb segmentPLOS ONE

Dear Dr. Callender,

Thank you for submitting your manuscript to PLOS ONE. After careful consideration, we feel that it has merit but does not fully meet PLOS ONE’s publication criteria as it currently stands. Therefore, we invite you to submit a revised version of the manuscript that addresses the points raised during the review process.

We look forward to receiving your revised manuscript.

Kind regards,

Haipeng Liu

Academic Editor

PLOS ONE

Journal Requirements:

Additional Editor Comments :

Dear Authors,

Thanks for the update. The early comments have been largely addressed, while some minor issues need further improvement. Please revise according to the reviewer's comments.

Reviewers' comments:

Reviewer's Responses to Questions

**Comments to the Author**

1. If the authors have adequately addressed your comments raised in a previous round of review and you feel that this manuscript is now acceptable for publication, you may indicate that here to bypass the “Comments to the Author” section, enter your conflict of interest statement in the “Confidential to Editor” section, and submit your "Accept" recommendation.

Reviewer #2: All comments have been addressed

Reviewer #4: (No Response)

Reviewer #5: All comments have been addressed

2. Is the manuscript technically sound, and do the data support the conclusions?

Reviewer #2: Yes

Reviewer #4: Yes

Reviewer #5: (No Response)

3. Has the statistical analysis been performed appropriately and rigorously? 

Reviewer #2: Yes

Reviewer #4: Yes

Reviewer #5: (No Response)

4. Have the authors made all data underlying the findings in their manuscript fully available?

Reviewer #2: Yes

Reviewer #4: (No Response)

Reviewer #5: (No Response)

5. Is the manuscript presented in an intelligible fashion and written in standard English?

Reviewer #2: Yes

Reviewer #4: Yes

Reviewer #5: (No Response)

6. Review Comments to the Author

Reviewer #2: (No Response)

Reviewer #4: Improvements in transitions, linking the literature review more closely to the study’s aims, and clarifying complex terminology would enhance its accessibility and impact.

The Methods section is thorough, but minor adjustments are needed in participant recruitment, data handling, and statistical analysis to ensure reproducibility.

The Results section presents robust, relevant data, but could be strengthened by improving data presentation clarity, providing more context, and discussing the physiological relevance in greater depth.

The discussion is solid, offering meaningful comparisons with existing literature and highlighting clinical implications, but would benefit from deeper mechanistic insights, more exploration of arterial data, and clearer articulation of clinical relevance.

These enhancements would position the study more effectively within vascular research and increase its potential impact on future clinical applications.

Reviewer #5: (No Response)

7. PLOS authors have the option to publish the peer review history of their article (what does this mean?). If published, this will include your full peer review and any attached files.

Reviewer #2: No

Reviewer #4: **Yes: **ERNEST MUSEKWA

Reviewer #5: No

---

## [Author Response · Author response to Decision Letter 1]

30 Oct 2024

Response to reviewers 29-10-24 (Second Revision)

External negative pressure transiently reduces intravenous pressure and augments the arteriovenous pressure gradient in the affected limb segment.

Many thanks for the further assistance with improving our manuscript. Responses to any new (second revision) comments are presented below.

Reviewer 1

Reviewer appears absent.

Reviewer 2

All previous comments addressed.

Reviewer 3

Reviewer appears absent.

Reviewer 4

Improvements in transitions, linking the literature review more closely to the study’s aims, and clarifying complex terminology would enhance its accessibility and impact.

The Methods section is thorough, but minor adjustments are needed in participant recruitment, data handling, and statistical analysis to ensure reproducibility.

The Results section presents robust, relevant data, but could be strengthened by improving data presentation clarity, providing more context, and discussing the physiological relevance in greater depth.

The discussion is solid, offering meaningful comparisons with existing literature and highlighting clinical implications, but would benefit from deeper mechanistic insights, more exploration of arterial data, and clearer articulation of clinical relevance.

These enhancements would position the study more effectively within vascular research and increase its potential impact on future clinical applications.

Thank you for these comments. We have addressed the above recommendations within the Revised Manuscript. Changes are highlighted in Red text and tracked throughout the document. Specifically, we have aimed to improve the flow and links between aims within the paper, as well as mechanistic discussion particularly in relation to the arterial measures. 

We stress that given our investigation of a group of healthy volunteers, we have made limited speculation as to the likely mechanisms underpinning any effects among clinical groups or patients with vascular disease. We have however tried to address this as best as possible given the lack of data specific to such patients, while avoiding unfounded speculation, or over-interpretation of our data.

Reviewer 5

No specific concerns or comments noted.

---

## [Decision Letter · Decision Letter 2]

22 Nov 2024

External negative pressure transiently reduces intravenous pressure and augments the arteriovenous pressure gradient in the affected limb segment

PONE-D-24-09979R2

Dear Dr. Callender,

We’re pleased to inform you that your manuscript has been judged scientifically suitable for publication and will be formally accepted for publication once it meets all outstanding technical requirements.

Kind regards,

Haipeng Liu

Academic Editor

PLOS ONE

Additional Editor Comments (optional):

Reviewers' comments:

Reviewer's Responses to Questions

**Comments to the Author**

1. If the authors have adequately addressed your comments raised in a previous round of review and you feel that this manuscript is now acceptable for publication, you may indicate that here to bypass the “Comments to the Author” section, enter your conflict of interest statement in the “Confidential to Editor” section, and submit your "Accept" recommendation.

Reviewer #4: (No Response)

Reviewer #6: All comments have been addressed

Reviewer #7: All comments have been addressed

2. Is the manuscript technically sound, and do the data support the conclusions?

Reviewer #4: (No Response)

Reviewer #6: Yes

Reviewer #7: Yes

3. Has the statistical analysis been performed appropriately and rigorously? 

Reviewer #4: (No Response)

Reviewer #6: Yes

Reviewer #7: Yes

4. Have the authors made all data underlying the findings in their manuscript fully available?

Reviewer #4: (No Response)

Reviewer #6: Yes

Reviewer #7: Yes

5. Is the manuscript presented in an intelligible fashion and written in standard English?

Reviewer #4: (No Response)

Reviewer #6: Yes

Reviewer #7: Yes

6. Review Comments to the Author

Reviewer #4: (No Response)

Reviewer #6: This is a high quality manuscript which has adequately addressed all previous revisions and offers a preliminary but insightful investigation into the role of INP for future clinical purposes. Research ethics and methodology are clearly illustrated highlighting rigour of the work provided.

Reviewer #7: Thank you for the opportunity to review the paper "External negative pressure transiently reduces intravenous pressure and augments the arteriovenous pressure gradient in the affected limb segment", presenting basic research about the effects of intermittent external negative pressure on intraarterial and intravenous blood pressure.

I have not reviewed this paper previously, therefore i cannot answer the question whether my previous comments have been addressed. From the R2 document it appears to me that the comments of the other reviewers have been addressed adequately, however the final decision about this is obviously up to the reviewer who made the comment.

Generally this study addresses a highly complex issue, with rather little previous knowledge, and has clear conclusions regarding the measured effects, which in my opinion merits publication. Introduction and methods section are thorough, and rather long, however, I consider this necessary given the complexity of the subject. The same appears for the results section.

Regarding the discussion, the authors have already addressed my major concern, that this study was performed in young, healthy participants, and not in vascular patients, and proposed further studies to investigate blood pressure changes in vascular patients. In my opinion, this is obviously necessary, maybe one should even compare patients where intermittent negative pressure therapy contributes to wound healing, and patients where this therapy does not succeed.

I have not found any minor issues to be corrected in the manuscript, and suggest that it is accepted for publication.

7. PLOS authors have the option to publish the peer review history of their article (what does this mean?). If published, this will include your full peer review and any attached files.

Reviewer #4: **Yes: **ERNEST MUSEKWA

Reviewer #6: **Yes: **Niraj S Kumar

Reviewer #7: **Yes: **Martin Altreuther

---

## [Editor Report · Acceptance letter]

26 Nov 2024

PONE-D-24-09979R2 

PLOS ONE

Dear Dr. Callender, 

I'm pleased to inform you that your manuscript has been deemed suitable for publication in PLOS ONE. Congratulations! Your manuscript is now being handed over to our production team.

Kind regards, 

on behalf of

Dr. Haipeng Liu 

Academic Editor

PLOS ONE